# Jet-Nemotron: Efficient Language Model with Post Neural Architecture Search

**Yuxian Gu, Qinghao Hu, Shang Yang, Haocheng Xi, Junyu Chen, Song Han, Han Cai***

NVIDIA

https://github.com/NVlabs/Jet-Nemotron

## Abstract

We present Jet-Nemotron, a new family of hybrid-architecture language models, which matches or exceeds the accuracy of leading full-attention models while significantly improving generation throughput. Jet-Nemotron is developed using Post Neural Architecture Search (PostNAS), a novel neural architecture exploration pipeline that enables efficient model design. Unlike prior approaches, PostNAS begins with a pre-trained full-attention model and freezes its MLP weights, allowing efficient exploration of attention block designs. The pipeline includes four key components: (1) learning optimal full-attention layer placement and elimination, (2) linear attention block selection, (3) designing new attention blocks, and (4) performing hardware-aware hyperparameter search. Our Jet-Nemotron-2B model achieves comparable or superior accuracy to Qwen3, Qwen2.5, Gemma3, and Llama3.2 across a comprehensive suite of benchmarks while delivering up to $53.6\times$ generation throughput speedup and $6.1\times$ prefilling speedup. It also achieves higher accuracy on MMLU and MMLU-Pro than recent advanced MoE full-attention models, such as DeepSeek-V3-Small and Moonlight, despite their larger scale with 15B total and 2.2B activated parameters.

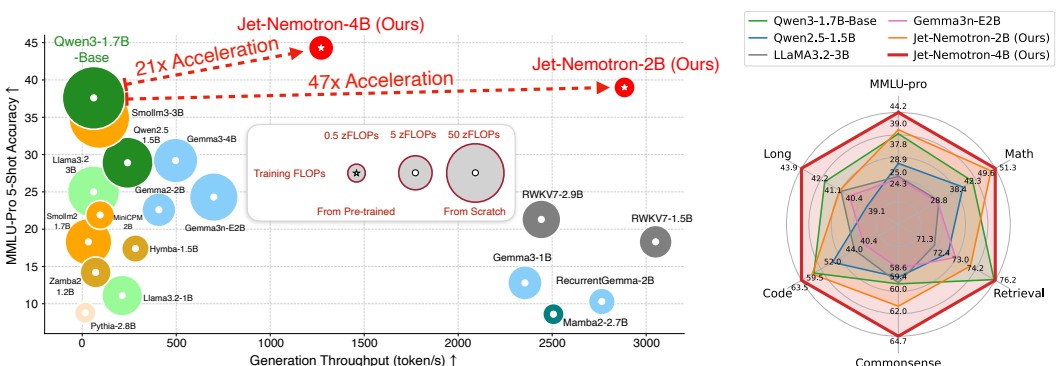

Figure 1: **Comparison Between Jet-Nemotron and State-of-the-Art Efficient Language Models.** The generation throughput is measured on the NVIDIA H100 GPU under a context length of 64K tokens. Jet-Nemotron-2B delivers a higher accuracy than Qwen3-1.7B-Base on MMLU-Pro while achieving $47\times$ higher generation throughput. Jet-Nemotron-4B, despite its larger model size, still achieves higher generation throughput than all full-attention models with less than 2B parameters.

## 1 Introduction

The rapid rise of Language Models (LMs) [1, 2, 3, 4, 5, 6, 7] marks a transformative era in artificial intelligence, with these models demonstrating exceptional accuracy across a broad range of tasks.

---

*Corresponding author(s): Han Cai (hcai@nvidia.com).

39th Conference on Neural Information Processing Systems (NeurIPS 2025).

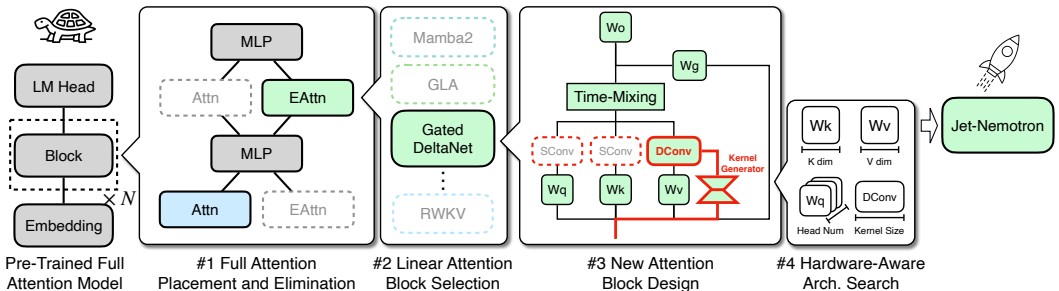

Figure 2: **PostNAS Roadmap.** Our pipeline starts from a pre-trained full-attention model and keeps the MLP frozen. It then performs a coarse-to-fine search for efficient attention block designs, first determining the optimal placement of full-attention layers, then selecting the best linear attention block or using a new linear attention block, and finally searching for optimal architectural hyperparameters.

However, their efficiency has become a significant concern due to the substantial computational and memory demands they impose. This issue is particularly pronounced in long-context generation and reasoning, where the self-attention mechanism [8] incurs a computational complexity of $O(n^2)$ and generates a large Key-Value (KV) cache[2].

To address this challenge, substantial efforts have been dedicated to designing more efficient LM architectures by developing attention mechanisms with reduced $O(n)$ complexity [9, 10, 11, 12, 13, 14]. In parallel, significant work has focused on constructing hybrid models that combine full and linear attention to strike a balance between accuracy and efficiency [15, 16, 17]. While these models offer improved efficiency, their accuracy still significantly falls behind state-of-the-art (SOTA) full-attention models, particularly on challenging benchmarks such as MMLU [18, 19], mathematical reasoning [20, 21, 22], retrieval [23, 24, 25], coding [26, 27, 28], and long-context tasks [29].

This paper introduces Jet-Nemotron, a new family of LMs that matches the accuracy of SOTA full-attention models while delivering exceptional efficiency. Figure 1 compares Jet-Nemotron with previous efficient LMs. Notably, Jet-Nemotron-2B achieves higher accuracy on MMLU-Pro than Qwen3-1.7B-Base [5], while offering $47\times$ higher generation throughput on the NVIDIA H100 GPU under a context length of 64K.

Jet-Nemotron is built upon Post Neural Architecture Search (PostNAS), a novel neural architecture exploration pipeline (Figure 2) that enables the rapid design of efficient model architectures. Unlike the mainstream LM architecture design approaches, PostNAS begins with a pre-trained full-attention model, from which it inherits the Multi-Layer Perceptron (MLP) weights and keeps them frozen throughout the process. This strategy significantly reduces training costs while still allowing for comprehensive exploration of the attention block. The pipeline then proceeds through four key steps to systematically search for optimal attention block designs.

**i) Full Attention Placement and Elimination.** Retaining a few full-attention layers within the model [30] is essential for maintaining high accuracy on challenging tasks such as retrieval. However, the optimal placement of these layers remains unclear. In Section 2.2, we introduce a novel approach that automatically learns where to use full-attention layers by training a once-for-all super network [31] (Figure 4). The resulting learned placement significantly outperforms the commonly used uniform placement strategy in terms of accuracy on MMLU (Figure 5, right).

**ii) Linear Attention Block Selection.** After finalizing the placement of full-attention layers, we conduct an attention block search to identify the optimal linear attention block (Section 2.3). Thanks to the low training cost of our framework, we can systematically evaluate existing linear attention blocks in terms of accuracy across diverse tasks, training efficiency, and inference speed. Importantly, our approach eliminates the need to rely on small proxy tasks, such as training tiny LMs (e.g., 50M or 150M parameters), ensuring that the search results directly translate to improvements in final model accuracy. Moreover, as new linear attention blocks are out, our framework can rapidly evaluate them against prior designs and adopt them if they demonstrate promising results.

---

[2]We refer to the standard $O(n^2)$ attention as full attention, and $O(n)$ attention as linear attention.

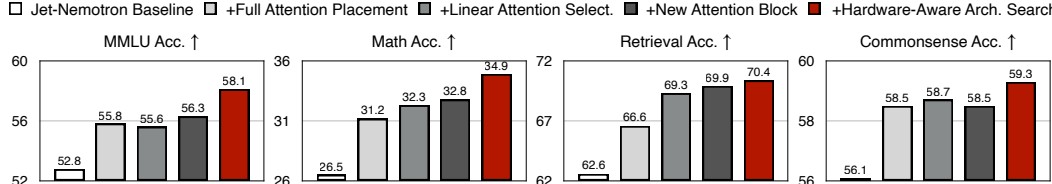

Figure 3: **PostNAS Accuracy Improvement Breakdown.** By applying PostNAS to the baseline model, we achieve significant accuracy improvements across all benchmarks.

**iii) New Attention Block Design.** PostNAS also facilitates the rapid design of new attention blocks. Adding convolutions is a widely used strategy to enhance the capacity of linear attention [32]. However, prior methods rely solely on static convolution kernels, lacking the ability to dynamically adapt convolution kernels' feature extraction patterns. In Section 2.4, we introduce a new linear attention block, JetBlock (Figure 2, #3). JetBlock uses a kernel generator to produce dynamic convolution kernels conditioned on the inputs, which are then applied to the value (V) tokens. Additionally, it removes redundant static convolutions on the query (Q) and key (K). Compared to prior linear attention blocks, JetBlock shows improved accuracy with a small overhead (Table 1).

**iv) Hardware-Aware Architecture Search.** Last, in Section 2.5, we introduce a hardware-aware architecture search to identify optimal architectural hyperparameters. Traditionally, the number of parameters has been used as a proxy for LM efficiency. However, parameter count does not directly correlate with generation efficiency on actual hardware. Our hardware-aware search discovers architectural hyperparameters that deliver similar generation throughput, while using more parameters to achieve better accuracy (Table 2).

We evaluate Jet-Nemotron across a comprehensive suite of benchmarks, including MMLU(-Pro) [18, 19], commonsense reasoning [33, 34, 35, 36, 37, 38], mathematical reasoning [20, 21, 22, 39], retrieval [23, 24, 25], coding [26, 27, 28, 40], and long-context tasks [29]. Our Jet-Nemotron-2B model matches or surpasses SOTA full-attention models, such as Qwen2.5 [4], Qwen3 [5], Gemma3 [41, 42] and Llama3.2 [2], across all benchmarks, while achieving significantly higher generation throughput. Furthermore, the throughput gains are even more substantial in long-context settings (Figure 6). For example, with a 256K context length, Jet-Nemotron-2B delivers a 6.14× prefilling speedup and a 53.6× decoding speedup compared to Qwen3-1.7B-Base. We hope that our efficient LM family (Jet-Nemotron), our new linear attention block (JetBlock), and our architecture design pipeline (PostNAS) will benefit the community and accelerate the development and deployment of next-generation efficient LMs. We summarize our main contributions below:

- We introduce PostNAS, a novel model architecture exploration paradigm for LMs. By reusing pre-trained LLMs, PostNAS reduces the cost and risk associated with LLM architecture exploration, enabling faster and more efficient innovation in the architecture design of LMs.

- We offer novel insights into the architecture design of efficient LMs, such as the task-specific importance of attention layers and the finding that KV cache size is a more critical factor than parameter count for generation throughput.

- We introduce a novel linear attention block, JetBlock, which integrates linear attention with dynamic convolution and hardware-aware architecture search. It consistently delivers accuracy improvements over previous linear attention blocks while maintaining comparable efficiency.

- We introduce Jet-Nemotron, a novel hybrid-architecture LM family that achieves superior accuracy across a wide range of tasks and offers significantly higher generation throughput than prior SOTA full-attention models (e.g., Qwen2.5, Qwen3, Gemma3, and Llama3.2). With its strong accuracy and exceptional inference efficiency, Jet-Nemotron offers practical benefits for various applications requiring efficient LMs.

## 2 Method

### 2.1 PostNAS Motivation and Roadmap

Designing new language model architectures is challenging and risky due to the high cost of pre-training. Moreover, the significant gap in computational resources and training data makes it difficult

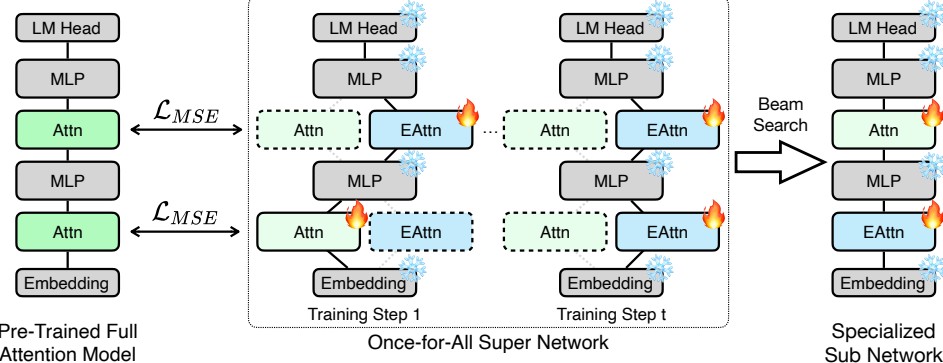

Figure 4: **Learning to Place Full Attention with PostNAS.** We train a once-for-all super network and perform beam search to identify the optimal placement of full attention layers.

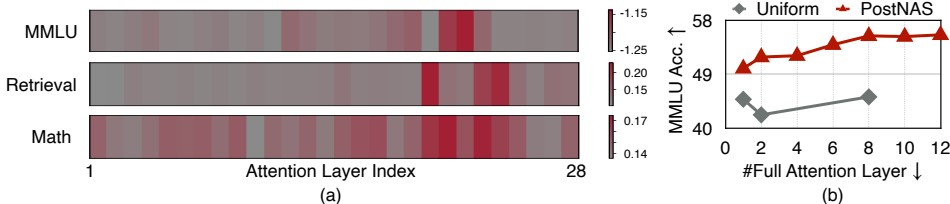

Figure 5: **(a) Layer Placement Search Results on Qwen2.5-1.5B.** Each grid cell represents the search objective value of the corresponding attention layer; higher values indicate greater importance. **(b) Comparison Between PostNAS and Uniform Placement.**

for researchers outside of major organizations to match the accuracy of state-of-the-art full-attention models developed by large industry players [4, 41, 2]. This disparity hinders innovation in language model architecture design.

This paper proposes an alternative strategy for developing new language model architectures. Rather than pre-training models from scratch, we explore novel architectures by building on top of existing full-attention models. This approach dramatically reduces both training costs and data requirements.

While architectures designed within this framework may not yield optimal results when trained from scratch, we argue that they remain highly valuable. First, as demonstrated in Figure 1, they can deliver immediate gains in efficiency and accuracy over state-of-the-art full-attention models, translating to practical benefits such as improved services and reduced operational costs. Second, our framework serves as a rapid testbed for architectural innovation. If a new design fails to perform well in this setting, it will be unlikely to succeed in full pre-training [43]. This filtering mechanism helps researchers avoid wasting substantial computational resources on unpromising designs.

Figure 2 illustrates the roadmap of PostNAS. Starting from a pre-trained full-attention model, it freezes the MLP weights and explores attention block designs in a coarse-to-fine manner through four key steps: full attention placement and elimination (Section 2.2), linear attention block selection (Section 2.3), new attention block design (Section 2.4), and hardware-aware architecture search (Section 2.5). Figure 3 shows the accuracy improvement breakdown from these steps. We observe substantial accuracy improvements across all benchmarks: +5.3 on MMLU, +8.4 on math, +7.8 on retrieval, and +3.2 on commonsense reasoning.

## 2.2 Full Attention Placement and Elimination

Incorporating a few full-attention layers has become a common strategy for improving accuracy [30, 16, 44, 17]. The standard approach applies full attention uniformly across a fixed subset of layers, with the remaining layers using linear attention. However, this uniform strategy is suboptimal, especially in our setting, where we begin with a pre-trained full-attention model.

To address this, we propose an automatic method for efficiently determining the placement of full-attention layers. The overall approach is illustrated in Figure 4. We construct a once-for-all super

network [45, 31] by augmenting the pre-trained full-attention model with alternative linear attention paths. During training, we randomly sample an active path at each step, forming a subnetwork, which is trained using feature distillation loss [46, 47, 48].

After training, we perform beam search [49] to determine the optimal placement of full-attention layers under a given constraint (e.g., two full-attention layers). The search objective is task-dependent: for MMLU, we select the configuration with the lowest loss on the correct answer (i.e., maximizing $-loss$), while for mathematical and retrieval tasks, we choose the one with the highest accuracy. As shown in Figure 5(b), PostNAS significantly outperforms uniform placement in terms of accuracy.

Figure 5(a) presents the search results for Qwen2.5-1.5B. For each layer, we extract the corresponding subnetwork from the super network by configuring that layer as full attention while setting all remaining layers to linear attention. We evaluate the accuracy or loss of each subnetwork on a given task and visualize the results using a heatmap. Our analysis reveals three key findings:

> **Key Finding 1:** In the pre-trained full-attention model, not all attention layers contribute equally. For MMLU, only two layers exhibit critical importance, while for retrieval tasks, just two to three layers are particularly crucial.

> **Key Finding 2:** Different attention layers contribute to different capabilities. Layers that are critical for MMLU accuracy are not necessarily important for retrieval tasks.

> **Key Finding 3:** The pattern of attention importance becomes more intricate for complex tasks like mathematical reasoning. Fortunately, the combined set of top critical layers identified for MMLU and retrieval already encompasses most of the key layers needed for math.

In addition to these key findings, we observe that the search results remain consistent when using different linear attention operations. In our final experiments, we use GLA [11] in the once-for-all super network training for simplicity and slightly improved training throughput.

## 2.3 Linear Attention Block Selection

Building on the discovered full-attention layer placement, we conduct an attention block search to identify the most suitable linear attention block for our setup. In our experiments, we evaluate six SOTA linear attention blocks, including RWKV7 [10], RetNet [12], Mamba2 [50], GLA [11], Deltanet [51], and Gated DeltaNet [32].

After initial efficiency profiling, we observe that RWKV7 exhibits significantly lower training throughput compared to other linear attention blocks, possibly due to suboptimal kernel implementation. Consequently, we exclude it from our training experiments. The results, summarized in Table 1, indicate that Gated DeltaNet achieves the best overall accuracy among the evaluated linear attention blocks. This is attributed to the combination of two factors: (1) the Data-Dependent Gating Mechanism [52], which dynamically controls whether the model should focus more on the current token or the history state, and (2) the Delta Rule [53], which updates the history state with the information increment from the current token, to save the limited state memory. Therefore, we proceed with Gated DeltaNet in our experiments.

## 2.4 New Attention Block Design

We propose a new linear attention block, JetBlock, designed to enhance the model's expressive power by incorporating dynamic convolution [54, 55] into linear attention. Convolution has been shown to be essential for achieving strong accuracy in many linear attention blocks [32, 56]. However, prior works typically use static convolution kernels, which cannot adapt their feature extraction patterns based on the input.

To address this limitation, we introduce a kernel generator module that dynamically produces convolution kernels based on the input features. The overall structure is shown in Figure 2 (#3). This module shares the same input as the Q/K/V projection layer and begins with a linear reduction

| Attention Block | Data-Depend Gating | Delta Rule | Throughput ↑ | | Accuracy ↑ | | | |
|---|:---:|:---:|---|---|---|---|---|---|
| | | | Training | Inference | MMLU | Math | Retreival | Common. |
| RWKV7 [10] | ✓ | ✓ | 123 | 2,542 | - | - | - | - |
| RetNet [12] | | | 269 | 2,535 | 53.6 | 29.9 | 63.7 | 58.1 |
| Mamba2 [50] | | | 273 | 3,220 | 51.5 | 26.0 | 68.9 | 57.5 |
| GLA [11] | ✓ | | 265 | 3,079 | 55.8 | 31.2 | 66.6 | 58.5 |
| Deltanet [51] | | ✓ | 254 | 2,955 | 48.9 | 27.4 | 67.9 | 56.6 |
| Gated DeltaNet [32] | ✓ | ✓ | 247 | 2,980 | 55.6 | 32.3 | 69.3 | 58.7 |
| JetBlock | ✓ | ✓ | 233 | 2,885 | 56.3 | 32.8 | 69.9 | 58.5 |
| + Hardware-Aware Search | ✓ | ✓ | 227 | 2,883 | **58.1** | **34.9** | **70.4** | **59.5** |

Table 1: **Accuracy and Efficiency of JetBlock**. JetBlock is designed through **Linear Attention Block Selection**, **New Attention Block Design**, and **Hardware-Aware Search**. It achieves higher accuracy than previous linear attention blocks while has comparable training and inference efficiency.

| $d_K$ | $d_V$ | $n_{head}$ | Params (B) | Cache Size (MB) | Throughput (token/s) ↑ | Retrieval Accuracy ↑ | Math Accuracy ↑ |
|---|---|---|---|---|---|---|---|
| 256 | 288 | 4 | 1.62 | 154 | 2,969 | 67.6 | 31.3 |
| 192 | 384 | 4 | 1.64 | 154 | 2,961 | 69.3 | 32.3 |
| 128 | 576 | 4 | 1.70 | 154 | 2,979 | 69.5 | 32.5 |
| 256 | 144 | 8 | 1.66 | 154 | 2,986 | 68.3 | 32.1 |
| 192 | 192 | 8 | 1.70 | 154 | 2,970 | 70.6 | 32.8 |
| 128 | 288 | 8 | 1.74 | 154 | 2,971 | 69.6 | 33.2 |
| 128 | 192 | 12 | 1.78 | 154 | 2,959 | 68.8 | 32.9 |
| 96 | 256 | 12 | 1.84 | 154 | 2,955 | 69.6 | 34.8 |
| 64 | 384 | 12 | 1.98 | 154 | 2,952 | 70.1 | 34.2 |

Table 2: **Detailed Results of Hardware-Aware Architecture Search.** The gray row is the original design [32], while the blue row shows the design produced by hardware-aware architecture search.

layer to improve efficiency, using a reduction ratio of 8. A SiLU activation function [57] is applied, followed by a final linear layer that outputs the convolution kernel weights. We adopt Gated DeltaNet for time-mixing, as it performs best compared with other designs as discussed in Section 2.3.

We apply the dynamic convolution kernels to the value (V) tokens, as applying them to the query (Q) or key (K) tokens offers little benefit. Furthermore, we find that static convolutions on Q and K can be removed with negligible impact on the final model accuracy once dynamic convolution is applied to V. We adopt this design in our final experiments for its slightly improved efficiency. Table 1 compares JetBlock with previous linear attention blocks. It provides better accuracy on math reasoning and retrieval tasks than Gated DeltaNet while maintaining similar efficiency.

## 2.5 Hardware-Aware Architecture Search

After finalizing the macro architecture, specifically the placement of full-attention layers, and selecting the linear attention block, we perform a hardware-aware architecture search to optimize core architectural hyperparameters, including key/value dimension and the number of attention heads.

Conventionally, parameter size is the primary efficiency metric used to guide model architecture design. However, this approach is suboptimal, as parameter count does not directly correlate with hardware efficiency. We address this limitation by using the generation throughput as a direct target for selecting architectural hyperparameters. We find that:

> **Key Finding 4:** KV cache size is the most critical factor influencing long-context and long-generation throughput. When the KV cache size is constant, models with different parameter counts exhibit similar generation throughput (Table 2).

This is because the decoding stage is typically memory-bandwidth-bound rather than compute-bound. In long-context scenarios, the KV cache often consumes more memory than the model weights.

| Type | Model | Params (B) | Cache Size (MB) | Throughput (token/s) ↑ | MMLU Acc. ↑ | MMLU-Pro Acc. ↑ | BBH Acc. ↑ |
|---|---|---|---|---|---|---|---|
| $O(n^2)$ | Qwen2.5-1.5B [4] | 1.5 | 1,792 | 241 | 59.5 | 28.9 | 44.1 |
| | Qwen3-1.7B-Base [5] | 1.7 | 7,168 | 61 | 60.3 | 37.8 | 54.2 |
| | Llama3.2-3B [2] | 3.0 | 7,168 | 60 | 54.9 | 25.0 | 47.1 |
| | MiniCPM-2B-128K [58] | 2.8 | 23,040 | 18 | 46.0 | 18.0 | 36.5 |
| | MobileLLM-1.5B [59] | 1.5 | 4,320 | 101 | 26.0 | 9.4 | 27.2 |
| | Smollm2-1.7B [60] | 1.7 | 12,288 | 32 | 48.5 | 18.3 | 35.1 |
| | DeepSeek-V3-Small@1.3T [6] | 2.2/15 | - | - | 53.3 | - | - |
| | Moonlight@1.2T [61] | 2.2/15 | - | - | 60.4 | 28.1 | 43.2 |
| $O(n)$ | Mamba2-2.7B [50] | 2.7 | 80 | 2,507 | 25.1 | 8.6 | 25.7 |
| | RWKV7-1.5B [10] | 1.5 | 24 | 3,050 | 41.0 | 13.4 | 15.9 |
| | Rec.Gemma-2B [62] | 2.0 | 16 | 2,355 | 28.6 | 12.8 | 33.3 |
| Hybrid | Gemma3n-E2B [42] | 2.0 | 768 | 701 | 53.9 | 24.3 | 45.1 |
| | Hymba-1.5B [44] | 1.5 | 240 | 180 | 49.7 | 17.4 | 29.8 |
| | Zamba2-1.2B [16] | 1.2 | 6,114 | 71 | 43.1 | 14.2 | 19.6 |
| | **Jet-Nemotron-2B** | 2.0 | 154 | 2,885 | 60.8 | 39.0 | 58.3 |
| | **Jet-Nemotron-4B** | 4.0 | 258 | 1,271 | **65.2** | **44.2** | **65.0** |

Table 3: **Results on MMLU and BBH.** DeepSeek-V3-Small@1.3T and Moonlight@1.2T are MoE models with 2.2B activated and 15B total parameters, trained on 1.3T and 1.2T tokens, respectively.

| Type | Model | Throughput (token/s) ↑ | Accuracy ↑ | | | | |
|---|---|---|---|---|---|---|---|
| | | | Avg. | GSM8K | MATH | MathQA | MMLU-Stem | GPQA |
| $O(n^2)$ | Qwen2.5-1.5B [4] | 241 | 38.4 | 62.4 | 13.1 | 34.4 | 52.7 | 29.4 |
| | Qwen3-1.7B-Base [5] | 61 | 42.3 | 62.8 | 16.7 | 46.0 | 50.8 | 27.9 |
| | Llama3.2-3B [2] | 60 | 28.8 | 25.8 | 8.6 | 34.2 | 45.3 | 30.1 |
| | MiniCPM-2B-128K [58] | 18 | 27.6 | 39.2 | 5.9 | 28.5 | 36.3 | 28.1 |
| | Smollm2-1.7B [60] | 32 | 28.9 | 30.3 | 9.2 | 33.7 | 41.3 | 30.1 |
| $O(n)$ | Mamba2-2.7B [50] | 2,507 | 16.6 | 3.0 | 3.9 | 24.3 | 26.6 | 25.3 |
| | RWKV7-1.5B [10] | 2,669 | 18.3 | 5.6 | 0.8 | 27.2 | 34.9 | 23.0 |
| | Rec.Gemma-2B [62] | 2,355 | 20.8 | 13.9 | 7.6 | 25.3 | 28.5 | 28.6 |
| Hybrid | Gemma3n-E2B [42] | 701 | 28.3 | 24.9 | 10.1 | 31.1 | 45.7 | 31.8 |
| | Hymba-1.5B [44] | 180 | 23.1 | 17.9 | 0.8 | 28.0 | 40.9 | 27.9 |
| | Zamba2-1.2B [16] | 71 | 24.8 | 28.1 | 5.9 | 26.0 | 36.5 | 27.7 |
| | **Jet-Nemotron-2B** | 2,885 | 49.6 | 76.2 | 23.3 | **53.8** | 62.7 | 32.1 |
| | **Jet-Nemotron-4B** | 1,271 | **51.3** | **78.7** | 25.2 | 52.5 | **65.6** | **34.6** |

Table 4: **Results on Math Tasks.**

Reducing its size decreases memory transfer time per decoding step and enables a larger batch size, thereby improving the generation throughput.

Based on Finding 4, we fix the KV cache size to match the original design and conduct a grid search over the key dimension, value dimension, and number of attention heads. Table 2 summarizes the results, where all variants use the same linear attention block (i.e., Gated DeltaNet) but have different configurations. The blue and gray rows represent our final design and the original one, respectively. Our final design achieves a generation throughput comparable to the original while incorporating more parameters and improving accuracy. From Table 1, we can see that hardware-aware search in PostNAS boosts the JetBlock's accuracy, while maintaining training and inference throughput.

## 3 Experiments

### 3.1 Setup

**Jet-Nemotron Model Family.** We construct two versions of Jet-Nemotron with different parameter sizes: Jet-Nemotron-2B and Jet-Nemotron-4B. We use the Retrieval task to guide the placement of

full attention layers and the MMLU task to guide the placement of sliding window attention (SWA) layers. Jet-Nemotron-2B is built upon Qwen2.5-1.5B [4], incorporating two full-attention layers (No. 15 and 20) for retrieval tasks and two sliding window attention (SWA) layers (No. 21 and 22) for multiple-choice tasks like MMLU. We find multiple-choice tasks mainly rely on the pattern-matching property of the softmax operation to route the knowledge of answers to their options. SWA effectively preserves the accuracy on such tasks. The remaining attention layers are replaced with JetBlock. Similarly, Jet-Nemotron-4B is based on Qwen2.5-3B and includes three full-attention layers (No. 18, 21, 33) and seven SWA layers (No. 6, 17, 20, 22, 23, 26, and 28). We summarize the final model architectures in Appendix A.1.

**Training Details.** The training consists of two stages. In the first stage, we freeze the MLPs and train the model using a distillation loss. In the second stage, we perform full-model training. At the first stage, we use a combination of Nemotron-CC [63] and Redstone-QA [64] as our pre-training corpus and train Jet-Nemotron models for 50B tokens. This is also the setting in Section 2 where we perform PostNAS. At the second stage, we include more high-quality data from math [65] and coding [66, 67] domains into our data mixture. The models are then trained on 350B tokens. We summarize the experimental costs in Appendix A.2.

**Evaluation Details.** We evaluate Jet-Nemotron across mainstream benchmark settings: MMLU(-Pro) [18, 19], mathematical reasoning [18, 20, 21, 22], commonsense reasoning [33, 34, 35, 36, 37, 38], retrieval [23, 24, 25], coding [26, 27, 28, 40], and long-context tasks [29]. We compare our models against state-of-the-art full-attention models [2, 4, 5], linear attention models [10, 50], and hybrid models [41, 44]. We adopt 4-shot evaluation for GSM8K [22] and MATH [18] and 5-shot evaluation for GPQA [20] and MMLU-Pro [19]. We use the official implementation of EvalPlus [40] and CRUXEval [28] for coding tasks. For all other tasks, we use the zero-shot setting. All evaluations are based on LM-Evaluation-Harness [68].

**Throughput Testbed.** Our throughput evaluation was performed on a DGX H100 server, featuring 8 NVIDIA H100 GPUs, 2 Intel Xeon Platinum 8480C (112 cores) CPUs, and 2TB of RAM. For fair and consistent comparisons, we employ the latest available software versions. Specifically, our environment include Pytorch 2.7.0 and Triton 3.3.0. We implement the full-attention block with FlashAttention 2.7.4 [69] and linear attention blocks with Flash-Linear-Attention 0.2.1 [70]. Model inference is based on the Transformers 4.52.0 implementation [71]. The context length is 64K, except stated explicitly, and each model is tested on a single H100 GPU. We report the cache sizes for a 64K input context in Table 3. When testing the throughput, we adopt chunk-prefilling [72] and search for the chunk sizes to maximize the batch size for each model under the constraint of the GPU memory. In this way, we measure the highest achievable decoding throughput on the device. We list the batch sizes used for each model in Appendix A.3.

## 3.2 Main Results on Accuracy

**Results on MMLU(-Pro) and BBH.** Table 3 compares Jet-Nemotron with the most advanced efficient language models. Jet-Nemotron-2B achieves 47× higher throughput and has 47× smaller cache size than Qwen3-1.7B-Base, while delivering significantly better accuracy on MMLU, MMLU-Pro, and BBH. Jet-Nemotron-2B even outperforms recent MoE models like DeepSeek-V3-Small [6] and Moonlight [61] with larger activated parameters (2.2B) and much larger total parameters (15B). When scaled to 4B parameters, Jet-Nemotron-4B *still maintains a 21× throughput advantage against Qwen3-1.7B-Base*. Compared to other linear attention and hybrid models, Jet-Nemotron also achieves substantially higher accuracy.

**Results on Math Tasks.** Table 4 reports our results on math tasks. Jet-Nemotron-2B achieves an average accuracy of 49.6, surpassing Qwen3-1.7B-Base by 6.3 while being 47× faster. In contrast, prior linear attention and hybrid models are far behind Qwen3 on math tasks.

**Results on Commonsense Reasoning Tasks.** Table 5 summarizes the results on commonsense reasoning tasks. Qwen2.5 and Qwen3 are relatively weak in this domain. Nevertheless, Jet-Nemotron-2B, which uses Qwen2.5-1.5B as the starting point, still demonstrates strong results, achieving an average accuracy of 62.0, outperforming all baseline models.

**Results on Long-Context Tasks.** A common concern with linear and hybrid architectures is their accuracy on long-context tasks. In Table 6, we evaluate this on LongBench [29] up to a 64K context length. Our findings show that Jet-Nemotron-2B, with two full-attention layers, achieves

| Model | Throughput (token/s) ↑ | Avg. | ARC-c | ARC-e | PIQA | Wino. | OBQA | BoolQ | TruthQA |
|---|---|---|---|---|---|---|---|---|---|
| Qwen2.5-1.5B [4] | 241 | 59.4 | 45.4 | 71.2 | 75.8 | 63.8 | 40.2 | 72.8 | 46.6 |
| Qwen3-1.7B-Base [5] | 61 | 60.0 | 44.9 | 68.6 | 75.5 | 63.8 | 39.0 | 79.0 | **48.8** |
| Llama3.2-3B [2] | 60 | 59.9 | 46.6 | 72.0 | 78.0 | 69.3 | 40.4 | 73.9 | 39.3 |
| MiniCPM-2B-128K [58] | 18 | 57.6 | 41.0 | 69.4 | 75.5 | 63.8 | 40.6 | 74.7 | 38.3 |
| Smollm2-1.7B [60] | 32 | 59.7 | 47.0 | 73.3 | 77.7 | 66.2 | 44.6 | 72.5 | 36.7 |
| Mamba2-2.7B [50] | 2,507 | 57.2 | 42.1 | 70.5 | 76.1 | 62.7 | 41.4 | 71.5 | 36.1 |
| RWKV7-1.5B [10] | 3,050 | 59.7 | 46.3 | 75.7 | 77.4 | 67.6 | **45.4** | 70.5 | 34.7 |
| Rec.Gemma-2B [62] | 2,355 | 46.5 | 29.4 | 41.5 | 66.6 | 54.1 | 27.0 | 72.0 | 34.7 |
| Gemma3n-E2B [42] | 701 | 58.6 | 43.2 | 73.1 | 77.0 | 60.8 | 40.8 | 76.0 | 39.1 |
| Hymba-1.5B [44] | 180 | 61.2 | 46.9 | 76.9 | 77.7 | 66.2 | 41.0 | 80.8 | 39.0 |
| Zamba2-1.2B [16] | 71 | 58.0 | 44.4 | 66.8 | 77.4 | 65.6 | 42.8 | 70.8 | 38.5 |
| **Jet-Nemotron-2B** | 2,885 | 62.0 | 48.6 | 74.8 | 75.4 | 65.8 | 40.6 | 81.2 | 47.8 |
| **Jet-Nemotron-4B** | 1,271 | **64.7** | **51.7** | **79.2** | **78.1** | **70.5** | 43.6 | **83.0** | 46.6 |

Table 5: **Results on Commonsense Tasks.**

| Type | Model | Throughput (token/s) ↑ | Avg. | Few-Shot | Code | Sum. | Single-Doc | Multi-Doc |
|---|---|---|---|---|---|---|---|---|
| $O(n^2)$ | Qwen2.5-1.5B [4] | 241 | 39.1 | 63.9 | 57.2 | 26.3 | 28.3 | 19.9 |
|  | Qwen3-1.7B-Base [5] | 61 | 42.2 | 68.8 | 48.1 | **26.8** | **36.6** | **30.6** |
|  | Llama3.2-3B [2] | 60 | 39.9 | 65.2 | 58.0 | 24.3 | 27.6 | 24.6 |
|  | MiniCPM-2B-128K [58] | 18 | 41.1 | 57.3 | 59.6 | 25.7 | 33.4 | 29.6 |
|  | Smollm2-1.7B [60] | 32 | 21.3 | 38.9 | 28.6 | 16.0 | 13.2 | 9.8 |
| $O(n)$ | Mamba2-2.7B [50] | 2,507 | 10.3 | 6.4 | 30.2 | 9.1 | 3.5 | 2.5 |
|  | RWKV7-1.5B [10] | 3,050 | 14.2 | 10.6 | 21.1 | 18.1 | 12.8 | 8.7 |
|  | Rec.Gemma-2.6B [62] | 2,355 | 24.1 | 31.8 | 56.7 | 12.9 | 9.2 | 9.6 |
| Hybrid | Gemma2-2.6B [73] | 388 | 22.9 | 28.7 | 52.0 | 12.6 | 13.9 | 7.3 |
|  | Gemma3n-E2B [73] | 701 | 40.4 | 56.4 | **67.2** | 25.6 | 29.3 | 28.6 |
|  | Hymba-1.5B [44] | 180 | 28.0 | 36.1 | 53.5 | 51.8 | 14.0 | 19.8 |
|  | Zamba2-1.2B [16] | 71 | 9.2 | 10.0 | 20.1 | 10.2 | 3.8 | 1.7 |
|  | **Jet-Nemotron-2B** | 2,885 | 41.1 | 68.7 | 58.1 | 26.0 | 30.8 | 21.9 |
|  | **Jet-Nemotron-4B** | 1,271 | **43.9** | **69.7** | 63.2 | 26.4 | 32.5 | 27.5 |

Table 6: **Results on Long-Context Tasks.**

performance comparable to leading models like Qwen2.5-1.5B and Gemma3n-E2B, which feature considerably more such layers. Furthermore, our Jet-Nemotron-4B outperforms Qwen3-1.7B-Base while delivering a 21× speedup in generation throughput. These results substantially advance the frontier of the efficiency-accuracy trade-off in long-context tasks.

**Results on Retrieval and Coding Tasks** We present the retrieval and coding results in Table 15 and Table 16 in Appendix B.4. On these tasks, Jet-Nemotron-2B performs comparably to Qwen3-1.7B-base. Jet-Nemotron-4B achieves a higher accuracy across all coding tasks while still delivering a large advantage on generation throughput against leading LMs like Qwen3-1.7B-Base.

**Summary.** Jet-Nemotron-2B and Jet-Nemotron-4B perform comparably with or even better than the advanced full-attention model (Qwen3-1.7B-Base) across all six evaluation domains. With significantly fewer full-attention layers and smaller KV cache size, Jet-Nemotron-2B and Jet-Nemotron-4B deliver 47× and 21× higher generation throughput than Qwen3-1.7B-Base, respectively.

## 3.3 Efficiency Benchmark Results

Figure 6 shows the throughput comparison between Qwen3-1.7B-Base and Jet-Nemotron-2B across various context lengths. During the prefilling stage, Jet-Nemotron-2B is initially 1.14 and 1.15 times faster than Qwen3-1.7B-Base at shorter context lengths (4K and 8K). This can be further improved by designing a better optimized kernel implementation of the JetBlock. As the context length increases,

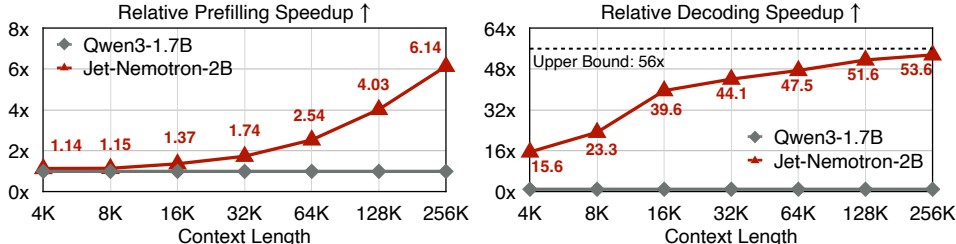

Figure 6: **Efficiency Comparison Across Different Context Lengths.** Jet-Nemotron-2B achieves up to a 6.14× speedup in prefilling and a 53.6× speedup in decoding compared to Qwen3-1.7B-Base.

the benefits of linear attention become prominent, making Jet-Nemotron-2B achieve 6.14× speedup at a 256K context length.

During the decoding stage, Jet-Nemotron-2B consistently outperforms Qwen3-1.7B-Base by a large margin. Since Jet-Nemotron-2B includes 2 full-attention layers with 2 groups of key-value states, its theoretical maximum speedup is $14 \times 4 = 56$ times compared to Qwen3-1.7B-Base with 28 full-attention layers, where each layer contains 8 groups of key-value states. In our throughput testbed, Jet-Nemotron-2B achieves a 15.6× speedup at a 4K context length and up to a 53.6× speedup at a 256K context length, almost reaching the theoretical upper bound.

## 4 Related Work

Large language models (LLMs) are powerful but computationally intensive, motivating many works to build efficient model architectures for LLMs. One line of research focuses on designing efficient linear attention blocks [9, 10, 11, 12, 32, 50, 51, 74, 75, 76, 77, 78, 79, 80, 81, 82, 83, 82, 84] or log-linear attention [85] blocks to replace full attention blocks. Orthogonally, another line of research tries to combine full attention and linear attention to build hybrid models [13, 15, 16, 17, 44, 86, 87, 88]. These works typically focus on the pre-training setting, and their accuracy lags behind leading full-attention models. Recently, there are some efforts on linearizing LLMs with full attention replaced with linear attention [89, 90, 91, 92, 93, 94, 95, 96]. However, their model architecture are poorly optimized due to the large overhead of evaluating specific configuration, and thus their results are still inferior to SOTA full-attention models.

Our work is also related to neural architecture search (NAS) [45, 97, 98, 99, 100], a powerful technique for exploring the architectural design space and discovering novel model structures. In particular, hardware-aware neural architecture search [45] enables the development of specialized model architectures optimized for target hardware by training a once-for-all super-network [31], or leveraging layer-wise distillation [101, 102], etc. However, NAS has been rarely applied in the era of large language models (LLMs) due to the prohibitive cost of pretraining. Recent efforts have primarily focused on building flexible LLM architectures [103, 104], which can generate a range of subnetworks with varying depths and widths to accommodate different hardware platforms. Nevertheless, the architectural backbone of these subnetworks remains unchanged, relying entirely on full-attention layers.

## 5 Conclusion

We introduce Jet-Nemotron, a new family of hybrid-architecture language models that outperform state-of-the-art full-attention models — including Qwen3, Qwen2.5, Gemma3, and Llama3.2 — while delivering substantial efficiency gains, with up to 53.6× higher generation throughput on H100 GPUs (256K context length, maximum batch size). Jet-Nemotron is enabled by two key innovations: (1) Post Neural Architecture Search, a highly efficient post-training architecture adaptation pipeline applicable to any pre-trained Transformer model; and (2) the JetBlock, a novel linear attention block that significantly outperforms prior designs such as Mamba2, GLA, and Gated DeltaNet. Extensive empirical results show that Jet-Nemotron achieves major efficiency improvements without compromising accuracy across a broad range of benchmarks. Additionally, Jet-Nemotron significantly reduces the cost and risk associated with LLM architecture exploration, enabling faster and more efficient innovation in language model design.

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

# A  Experimental Details

## A.1  Final Model Architecture

The final Jet-Nemotron models are composed of a stack of blocks, each containing a Multi-Layer Perceptron (MLP) layer and an attention layer. The attention layer is selected from one of three types: full attention, sliding window attention, or JetBlock. The detailed architecture configurations are presented in Table 7.

|                                | Jet-Nemotron-2B | Jet-Nemotron-4B         |
|--------------------------------|-----------------|-------------------------|
| Total blocks                   | 28              | 36                      |
| Full Attention Layers          | No. 15, 20      | No. 18, 21, 22, 28, 33  |
| Sliding Window Attention Layers| No. 21, 22      | No. 17, 20, 23, 24, 26  |
| Vocabulary Size                | 151,643         | 151,643                 |
| Hidden Size                    | 1,536           | 2,048                   |
| MLP Intermediate Size          | 8,960           | 11,008                  |

Table 7: The overall model architectures of Jet-Nemotron families.

The full attention and sliding window attention layers use grouped-query attention [105] and are configured as in Table 8. For sliding window attention layers, the window size is set to 1,152 in Jet-Nemotron-2B and 2,048 in Jet-Nemotron-4B.

| Full Attention / SWA   | Jet-Nemotron-2B | Jet-Nemotron-4B |
|------------------------|-----------------|-----------------|
| Attention Head Number  | 12              | 16              |
| Dimensions of Q/K/V    | 128             | 128             |
| K/V Head Number        | 2               | 2               |
| Position Embedding     | RoPE            | RoPE            |

Table 8: The configurations of full-attention layers in Jet-Nemotron models.

The configuration of JetBlock are shown in Table 9 :

| JetBlock                    | Jet-Nemotron-2B | Jet-Nemotron-4B |
|-----------------------------|-----------------|-----------------|
| Q/K Dimension               | 96              | 128             |
| V Dimension                 | 256             | 256             |
| Head Number                 | 12              | 16              |
| Convolution Kernel Size     | 4               | 4               |
| DConv Generator Hidden Size | 32              | 32              |

Table 9: The configurations of JetBlock.

## A.2  Experimental Costs

Table 10 summarizes the costs for PostNAS and training the Jet-Nemotron-2B model. We used 32 H100 GPUs in parallel. The reported GPU hours already account for the total number of devices.

## A.3  Throughput Measurement

Throughout the experiments, we measure the maximum reachable prefilling and decoding throughput of Jet-Nemotron and the baselines on a single H100 GPU. This is achieved by adjusting the chunk size in chunk-prefilling [72] to maximize the decoding batch size without sacrificing the prefilling throughput. We list the optimized batch size and the corresponding chunk size for each model in Table 11. The prefilling context length is 64K. Since the KV cache memory dominates GPU usage during inference, by reducing the memory footprint per sequence, smaller caches allow more sequences to be processed in parallel, greatly boosting generation throughput.

|  |  | Tokens (B) | ZFLOPs | Time (H100 GPU Hours) |
|---|---|---|---|---|
| PostNAS | Full Attention Placement and Elimination | 50 | 0.8 | 808 |
|  | Linear Attention Block Selection | 50 | 4.0 | 3120 |
|  | New Attention Block Design | 50 | 0.8 | 624 |
|  | Harware-Aware Arch Search | 50 | 7.2 | 5616 |
| Training | Stage1 | 50 | 0.8 | 624 |
|  | Stage2 | 350 | 5.6 | 7536 |

Table 10: Experimental Costs for PostNAS and training the Jet-Nemotron-2B model.

| Model | Batch Size | Chunk Size |
|---|---|---|
| Qwen2.5-1.5B | 32 | 8,192 |
| Qwen3-1.7B | 8 | 16,384 |
| Llama3.2-1B | 32 | 4,096 |
| MiniCPM-2B-128K | 2 | 2,048 |
| Pythia-2.8B | 2 | 16,384 |
| Smollm2-1.7B | 4 | 16,384 |
| Mamba2-2.7B | 128 | 1,024 |
| RWKV7-1.5B | 256 | 2,048 |
| Rec.Gemma-2B | 128 | 512 |
| Gemma3n-E2B | 64 | 4,096 |
| Gemma2-2.6B | 16 | 2,048 |
| Hymba-1.5B | 64 | 512 |
| Zamba2-1.2B | 8 | 8,192 |
| Jet-Nemotron-2B | 128 | 2,048 |
| Jet-Nemotron-4B | 64 | 1,024 |

Table 11: **Hyper-Parameters in Efficiency Measurement.** We adjust the chunk size to maximize decoding batch size without compromising prefilling throughput.

# B  Additional Results

## B.1  Controlled Study on Training Data

To exclude the influence of training data, we continually pre-train the baseline models (Qwen2.5, RWKV-7, and Mamba-2) on Jet-Nemotron 's training dataset to provide a more comprehensive evaluation. The results in Table 12 show that Jet-Nemotron-2B outperforms all these finetuned baseline models by a significant margin.

| Model | MMLU | Math | Commonsense | Retrieval |
|---|---|---|---|---|
| Qwem2.5-1.5B-continual | 56.7 | 37.6 | 59.8 | 71.5 |
| Mamba2-2.7B-continual | 41.0 | 22.5 | 56.9 | 55.9 |
| RWKV7-1.5B-continual | 49.8 | 25.2 | 59.3 | 57.2 |
| Jet-Nemotron-2B | **59.6** | **40.2** | **61.7** | **73.6** |

Table 12: **Controlled Study on Training Data**. All models are pre-trained or continually pre-trained on the Jet-Nemotron stage-2 training corpus discussed in Section 3.1.

## B.2  Throughput Results on Lower-End Hardware

We measure the throughput of Jet-Nemotron-2B and Qwen2.5-1.5B on the NVIDIA Jetson Orin (32GB) and NVIDIA RTX 3090 GPUs with a context length of 64K. Results in Table 13 show that Jet-Nemotron-2B achieves 8.84× and 6.50× speedups over Qwen2.5-1.5B on the Jetson Orin and RTX 3090 GPUs, respectively.

| Hardware | Qwen2.5-1.5B (Tokens/s) | Jet-Nemotron-2B (Tokens/s) | SpeedUp |
|---|---|---|---|
| Orin | 6.22 | 55.00 | 8.84 |
| 3090 | 105.18 | 684.01 | 6.50 |

Table 13: Throughput Results on Jetson Orin (32GB) and NVIDIA RTX 3090 GPUs.

| Model | Throughput | Accuracy ↑ | | | | | |
|---|---|---|---|---|---|---|---|
| | (token/s) ↑ | MMLU | MATH | Common. | Retrieval | Code | Long-Context |
| Falcon-H1-1.5B [106] | 223 | 60.5 | 40.1 | 59.9 | 73.5 | 56.0 | 40.7 |
| Falcon-H1-1.5B-deep [106] | 66 | 63.5 | 46.8 | 60.6 | 74.6 | 60.3 | 33.4 |
| **Jet-Nemotron-2B** | 2,885 | 60.8 | 49.6 | 62.0 | 74.2 | 59.5 | 41.1 |
| **Jet-Nemotron-4B** | 1,271 | **65.2** | **51.3** | **64.7** | **76.2** | **63.5** | **43.9** |

Table 14: **Comparison with Falcon-H1.**

## B.3 Comparison to Falcon-H1

We compare our work with the concurrent Falcon-H1 [106], a hybrid model that incorporates Mamba2 [50] and full attention. Unlike Jet-Nemotron, which alternates between component types at the layer level, Falcon-H1 employs a head-wise hybrid strategy. As shown in Table 14, Jet-Nemotron-2B outperforms Falcon-H1-1.5B and is comparable to Falcon-H1-1.5B-deep in accuracy, while achieving significantly higher generation throughput. Jet-Nemotron-4B outperforms both the two Falcon-H1 models while still achieves higher generation throughput. This efficiency gap arises because the head-wise strategy requires sequential computation of Mamba2 and full attention operations within a single layer, thereby limiting parallelism. The "-deep" variant further exacerbates this issue by reducing model width in favor of greater depth.

## B.4 Results on Retrieval and Coding

Table 15 and Table 16 presents the results on retrieval and tasks. Jet-Nemotron-2B outperforms all baselines except Qwen3-1.7B-Base. When scaled to 4B, Jet-Nemotron-4B achieves the best average accuracy of 76.2, while still maintaining 21× speedup compared to Qwen3.

| Type | Model | Throughput (token/s) ↑ | Accuracy ↑ | | | |
|------|-------|------------------------|-----|-----|------|-------|
| | | | Avg. | FDA | SWDE | Squad |
| $O(n^2)$ | Qwen2.5-1.5B [4] | 241 | 72.4 | **82.8** | 86.3 | 48.1 |
| | Qwen3-1.7B-Base [5] | 61 | 76.1 | 81.8 | 89.2 | 57.2 |
| | Llama3.2-3B [2] | 60 | 71.3 | 82.3 | 89.6 | 56.4 |
| | MiniCPM-2B-128K [58] | 18 | 72.6 | 72.3 | 86.4 | **59.1** |
| | Smollm2-1.7B [60] | 32 | 68.9 | 78.1 | 82.4 | 46.3 |
| $O(n)$ | Mamba2-2.7B [50] | 2,507 | 57.0 | 51.7 | 74.3 | 45.1 |
| | RWKV7-1.5B [10] | 3,050 | 58.6 | 54.5 | 73.3 | 48.0 |
| | Rec.Gemma-2.6B [62] | 2,355 | 68.8 | 62.3 | 86.4 | 57.8 |
| Hybrid | Gemma3n-E2B [73] | 701 | 74.0 | 77.3 | 86.4 | 58.2 |
| | Hymba-1.5B [44] | 180 | 57.1 | 46.6 | 74.4 | 50.2 |
| | Zamba2-1.2B [16] | 71 | 66.4 | 73.8 | 80.7 | 44.8 |
| | **Jet-Nemotron-2B** | 2,885 | 74.2 | 80.4 | 85.7 | 56.6 |
| | **Jet-Nemotron-4B** | 1,271 | **76.2** | 82.5 | **89.7** | 56.4 |

Table 15: **Results on Retrieval Tasks.**

| Type | Model | Throughput (token/s) ↑ | Accuracy ↑ | | | |
|------|-------|------------------------|-----|----------|---------------|---------------|
| | | | Avg. | EvalPlus | CRUXEval-I-cot | CRUXEval-O-cot |
| $O(n^2)$ | Qwen2.5-1.5B [4] | 241 | 52.0 | 54.3 | 56.0 | 45.8 |
| | Qwen3-1.7B-Base [5] | 61 | 58.9 | 62.8 | 60.4 | 53.4 |
| | Llama3.2-3B [2] | 60 | 44.0 | 35.5 | 54.7 | 41.7 |
| | MiniCPM-2B-128K [58] | 18 | 34.2 | 40.7 | 29.9 | 31.9 |
| | Smollm2-1.7B [60] | 32 | 36.2 | 20.6 | 49.5 | 38.6 |
| $O(n)$ | Mamba2-2.7B [50] | 2,507 | 14.0 | 12.0 | 9.3 | 20.7 |
| | RWKV7-1.5B [10] | 3,050 | 13.2 | 16.8 | 8.0 | 14.7 |
| | Rec.Gemma-2.6B [62] | 2,355 | 36.8 | 29.5 | 46.7 | 34.2 |
| Hybrid | Gemma3n-E2B [73] | 701 | 40.4 | 29.6 | 49.9 | 41.6 |
| | Hymba-1.5B [44] | 180 | 30.3 | 31.3 | 32.2 | 27.5 |
| | Zamba2-1.2B [16] | 71 | 20.1 | 12.7 | 21.1 | 26.4 |
| | **Jet-Nemotron-2B** | 2,885 | 59.5 | 60.8 | 61.1 | 56.7 |
| | **Jet-Nemotron-4B** | 1,271 | **63.5** | **65.6** | **65.9** | **59.0** |

Table 16: **Results on Coding Tasks.**

