# OpenReview forum: "Jet-Nemotron: Efficient Language Model with Post Neural Architecture Search"
_NeurIPS.cc/2025/Conference — NeurIPS 2025 poster_

### Official Review · Reviewer_VP6K · 2025-06-24

**Clarity:** 4
**Significance:** 3
**Originality:** 3
**Rating:** 5
**Confidence:** 5

**Summary:**

This paper proposes JetLM, a new method of designing LLMs with similar performance as SOTA at improved inference efficiency.

**Questions:**

Please see the weaknesses.

**Ethical Concerns:**

["NO or VERY MINOR ethics concerns only"]

**Final Justification:**

Overall, I enjoy this paper, and the idea of applying NAS to mobile-friendly LLMs is an important research question in both academia and industry. I had some initial concerns about the system-related evaluations, but the authors have addressed the majority of them in the rebuttal.

Therefore, I recommend accept for this paper.

**Limitations:**

Please see the weaknesses.

**Quality:**

3

**Strengths And Weaknesses:**

**Strengths**:

1. This paper is easy to follow with clear writing and presentations.
2. The idea of applying NAS to design LLMs, especially in the small model domain, is an interesting and promising direction.

**Weaknesses**:
1. The main concern I have with the paper is the throughput evaluation. The authors employ chunk-prefilling for long context tasks. I am wondering how would JetLM performs if using the same batch size and sequence length.
2. It would be better if the authors could compare JetLM to MobileLLM [1].
2. It would be better if the authors would test out the performance for JetLM on lower end hardware, like older generations of NVIDIA GPUs, or Jetson Nano and raspberry pi, since these machines would be the primary deployment platforms for small LLMs.
3. It would be better if the authors share more details on the training process. For instance, what is the total training time and how many tokens are used in pre-training?
4. Some related works on NAS for LLM are missing for reference [2-3].

[1] MobileLLM: Optimizing Sub-billion Parameter Language Models for On-Device Use Cases, ICML 2024.

[2] Search for Efficient Large Language Models, NeurIPS 2024.

[3] MeRino: Entropy-driven Design for Generative Language Models on IoT Devices, AAAI 2025.

---

> ### Author Rebuttal · Authors · 2025-07-31
>
> ### Q1: Throughput Comparison with Equal Batch Size and Sequence Length
> First, we would like to clarify that the throughput comparison using chunk-prefilling—where the batch size is maximized under a fixed GPU memory constraint—constitutes a fair, apples-to-apples evaluation. This setting reflects the maximum achievable throughput for each model.
>
> To provide a more comprehensive comparison, we also evaluate the throughput of JetLM-2B and Qwen2.5-1.5B on an H100 GPU using identical batch sizes and sequence lengths. The results are summarized below:
> |   Sequence Length   | Qwen2.5-1.5B (Tokens/s) | JetLM-2B (Tokens/s) | SpeedUp |
> | ----------------------- | ----------------------- | ------------------- | ------- |
> | 4K   | 3424                    | 13233               | 3.86    |
> | 8K   | 1830                    | 8462                | 4.62    |
> | 16K  | 926                     | 4631                | 5.00    |
> | 32K  | 490                     | 2536                | 5.17    |
> | 64K  | 241                     | 1295                | 5.38    |
> | 128K | 119                     | 656                 | 5.53    |
> | 256K | 61                      | 337                 | 5.55    |
>
> Under this setting, JetLM-2B delivers up to a 5.55× speedup compared to Qwen2.5-1.5B.
>
> ### Q2: Comparison with MobileLLM
> Thank you for bringing this relevant work to our attention. We will include comparisons with MobileLLM in the revised manuscript. The results are provided below:
>
> |       Model  | Throughput | MMLU | Math | Commonsense | Retrieval |
> | -------------- | ---------- | ---- | ---- | ----------- | --------- |
> | MobileLLM-1.5B | 101        | 26.0 | 16.0 | 50.6        | 54.0      |
> | JetLM-2B       | 2863       | 59.6 | 40.2 | 61.7        | 73.6      |
>
> JetLM-2B achieves higher throughput than MobileLLM-1.5B while also delivering superior accuracy on downstream tasks, including MMLU, Math, Commonsense, and Retrieval.
>
> ### Q3: Throughput Results on Lower-End Hardware
> We measure the throughput of JetLM-2B and Qwen2.5-1.5B on the NVIDIA Jetson Orin (32GB) and NVIDIA RTX 3090 GPUs with a context length of 64K:
>
> |   Hardware  | Qwen2.5-1.5B (Tokens/s) | JetLM-2B (Tokens/s) | SpeedUp |
> | ---- | ----------------------- | ------------------- | ------- |
> | Orin | 6.22                    | 55.00              | 8.84    |
> | 3090 | 105.18 	         | 684.01 	    | 6.50 |
>
> JetLM achieves 8.84x and 6.50x speedups over Qwen2.5 on the Jetson Orin and RTX 3090 GPUs, respectively.
>
> ### Q4: Training Details
> The total training time and number of tokens used to train JetLM-2B are summarized below:
>
> |        | Tokens | Time (H100 GPU Hours) |
> | ------ | ------ | ---- |
> | Stage1 | 50B    | 624  |
> | Stage2 | 100B   | 1344 |
>
> ### Q5: Related Work
> Thank you for pointing out these references. We will include and discuss them in the revised manuscript.

---

> > ### Comment · Reviewer_VP6K · 2025-07-31
> > **Response**
> >
> > Thank you for your detailed rebuttal. I have no further questions and will increase my score from 4 to 5.

---

### Official Review · Reviewer_8wVd · 2025-07-03

**Clarity:** 4
**Significance:** 3
**Originality:** 4
**Rating:** 4
**Confidence:** 3

**Summary:**

The authors propose JetLM, which is a neural architecture search framework for creating distilled hybrid-attention language models sourced from original pre-trained language models. They decide a final architecture by training a super-network that contains both full-attention and their novel sub-attention Gated Deltanet modules whose paths are randomly sampled during forward-passes. The optimal model architecture is then beam-searched via task-specific objectives after training. Afterwards, they lead a hyperparameter search to maximize throughput. The resulting architecture is then trained in stages using distillation and full-parameter finetuning. The resulting JetLM model has strong empirical performance on benchmarks in terms of accuracy and latency, although some questions remain regarding the method and evaluation.

**Questions:**

- "92: If a new design fails to perform well in this setting, it will be unlikely to succeed in full pre-training." Any evidence for this claim?
- What is the justification for the SWA layer placement? Why were those specific layer numbers chosen?
- Is the throughput of the different attention block types reliant on their individual kernel implementations? If so, is there some kind of minimal control experiment that can be run to understand how their throughput compares, irrespective of those implementations?
- If the method "enables the rapid design of efficient model architectures", then why wasn't it applied to a 7B model?
- Can you better explain how, after searching for the best attention placement over multiple tasks, the final configuration was chosen?

The evaluation weaknesses here raises some concerns for me. It is both unclear whether the attention block comparison is fair, due to potential differences in implementation optimization/experimental settings, and if linear attention models would have performed better than JetLM due to the finetuning data. These issues make it uncertain whether JetLM has an empirical advantage over linear-attention models. On the other hand, the results are promising and the method is both novel and useful. To that end, my score is a weak reject. If these weaknesses (and the following questions) can be adequately addressed, I would be happy to raise my score.

**Ethical Concerns:**

["NO or VERY MINOR ethics concerns only"]

**Final Justification:**

Many of the points from my original review stand on the novelty of this method. My concerns, of which there were many, were addressed satisfactorily by the authors. I encourage them to include the new results and explanations in their final paper.

**Limitations:**

- NAS method appears to require task-specific configuration (line 113).

**Quality:**

2

**Strengths And Weaknesses:**

Strengths:
- Novel insights regarding task-specific attention layer importance
- the method appropriately adapts to target hardware
- the throughput speedups are significant and impressive
- the efficiency of the method scales with context length
- the resulting JetLM model is evaluated on a wide suite of benchmarks
- the paper is very well-written

Weaknesses:
- In the study comparing different attention blocks, how do we know the comparison is fair? Was the number of new parameters held constant? Or perhaps the latency?
- The JetLM model is distilled and finetuned on a new dataset, whereas the pre-trained LLM baselines are not. This difference in experimental setup creates uncertainty -- would finetuning the O(n) baselines on the same dataset have yielded similar results to JetLM?
- There was no information provided on how long the experiments took.

---

> ### Author Rebuttal · Authors · 2025-07-31
>
> ### Q1: On the Study Comparing Different Attention Blocks
> In this study, we ensure that all models use the same cache size to guarantee fair comparisons. As discussed in Section 2.5, cache size is the primary factor affecting generation throughput in long-context settings.
>
> ### Q2: Finetuning Baseline Models on JetLM Training Dataset
> Following your advice, we conducted experiments to provide more comprehensive comparisons with the baseline models. The results are summarized below:
>
> |        Model      | MMLU | Math | Commonsense | Retrieval |
> | --------------------- | ---- | ---- | ----------- | --------- |
> | Qwem2.5-1.5B-Finetune | 56.7 | 37.6 | 59.8        | 71.5      |
> | Mamba2-2.7B-Finetune  | 41.0 | 22.5 | 56.9        | 55.9      |
> | RWKV7-1.5B-Finetune   | 49.8 | 25.2 | 59.3        | 57.2      |
> | JetLM-2B              | **59.6** | **40.2** | **61.7**        | **73.6**      |
>
> JetLM still provides a substantial accuracy improvement compared to the fine-tuned baseline models.
>
> ### Q3: Experiment Running Time
> The search and training times for JetLM are summarized below:
>
> |           |                        | Tokens (B) | ZFLOPs |  Time (H100 GPU Hours) |
> | --------- | ---------------------------------------- | ------ | -------- | -------- |
> | Search | Full Attention Placement and Elimination | 50 | 0.8    | 808    |
> |           | Linear Attention Block Selection         | 50       | 4.0    | 3120    |
> |           | New Attention Block Design               | 50       | 0.8    | 624      |
> |           | Harware-Aware Arch Search              | 50       | 7.2    | 5616    |
> | Training  | Stage1                                           | 50        | 0.8    | 624     |
> |           | Stage2                                                | 100      | 1.6    | 1344   |
>
>
>
> ### Q4: PostNAS as a Proxy Testbed for Architecture Exploration (L92)
> In the neural architecture search literature [1], leveraging results from a proxy task for architecture selection is a widely adopted strategy. This supports our conjecture that PostNAS can serve as an effective proxy task for architecture selection. We will revise the manuscript accordingly.
>
> ### Q5: SWA Layer Placement and Final Attention Placement Configuration
> We determine the attention placement configuration based on our super-network search results, as detailed in Sections 2.2 and 3.1. Specifically, we use the Retrieval task to guide the placement of full attention layers and the MMLU task to guide the placement of sliding window attention (SWA) layers. As a result, we assign full attention to the top two critical layers for Retrieval (layers 15 and 20), and SWA to the top two critical layers for MMLU (layers 21 and 22).
>
> ### Q6: Kernel Implementation
> Kernel implementations can impact throughput results. To ensure a fair comparison, we adopt the unified Triton implementations from flash-linear-attention [2] as a controlled experimental setting.
>
> ### Q7: Apply to a 7B Model
> While our method significantly reduces the cost of model architecture exploration, experiments with 7B-scale models still require substantial computational resources. We plan to open-source additional accelerated models with 7B+ parameters using our approach. However, due to hardware constraints, it is not feasible to complete these experiments within the rebuttal period. We will include the new results in the next version of our work.
>
> [1] Zoph, Barret, et al. “Learning transferable architectures for scalable image recognition.” Proceedings of the IEEE conference on computer vision and pattern recognition. 2018.
>
> [2] Yang, Songlin and Zhang, Yu. “FLA: A Triton-Based Library for Hardware-Efficient Implementations of Linear Attention Mechanism.” GitHub. 2024.

---

> > ### Comment · Reviewer_8wVd · 2025-08-01
> >
> > I thank the authors for their response, which clarified and addressed many of my questions. I have just two more, if they will kindly oblige:
> >
> > Regarding Q1, while it is certainly empirically true that cache size is the primary factor, is there any explanation as to why? This is a novel insight, and it would be nice for the manuscript to dive deeper into this.
> >
> > Regarding Q3, can you further describe the number of devices used concurrently to attain that number of GPU hours?

---

> > > ### Author Response · Authors · 2025-08-02
> > >
> > > It’s great to hear that our responses addressed your initial questions—thank you for the thoughtful follow-ups.
> > >
> > > ### Q1 Follow-up
> > >
> > > In long-context settings, using smaller cache sizes significantly improves throughput due to two compounding factors:
> > > - **Larger batch sizes**: KV cache memory dominates GPU usage during inference. By reducing the memory footprint per sequence, smaller caches allow more sequences to be processed in parallel, greatly boosting generation throughput.
> > > - **Lower memory transfer overhead**: The decoding stage is typically memory-bandwidth-bound rather than compute-bound. In long-context scenarios, the KV cache often consumes more memory than the model weights. Reducing its size decreases memory transfer time per decoding step, thereby reducing per-token latency.
> > >
> > > These two effects compound, making cache size the primary determinant of throughput in long-context inference. We will include a detailed discussion in our revised manuscript—thank you for the suggestion!
> > >
> > > ### Q3 Follow-up
> > >
> > > We used 32 H100 GPUs in parallel. The reported GPU hours already account for the total number of devices.

---

> > > > ### Comment · Reviewer_8wVd · 2025-08-03
> > > >
> > > > Thank you for addressing my concerns. I will raise my score accordingly. I highly encourage the authors to include the answer to these questions in their final manuscript.

---

> > > > > ### Author Response · Authors · 2025-08-04
> > > > >
> > > > > Thank you! We appreciate your feedback and will include the answers in the final manuscript.

---

### Official Review · Reviewer_Ndha · 2025-07-03

**Clarity:** 2
**Significance:** 3
**Originality:** 2
**Rating:** 4
**Confidence:** 3

**Summary:**

This article proposes JetLM based on post-neural architecture search. PostNAS is a new attention mechanism that improves generation throughput while retaining performance comparable to full attention. This article adopts full attention placement and linear attention module selection. Its linear attention module uses DynaConvGDN to dynamically generate convolution kernels based on input features and apply dynamic convolution kernels to values (V). It uses a fixed KV cache size and performs small-scale grid search on key dimensions, value dimensions, and number of attention heads.

**Questions:**

1. Need to compare with other SOTA models on MMLU, Math Task and Commonsense Task.
2. Or provide a control experiment that excludes the influence of training data.
3. Need to compare the difference between DynaConvGDN and Gated DeltaNet in Benchmark.
4. Provide the reasons for using full attention layer in 15, 20, SWA layer in 21, 22, and linear attention layer in the rest

**Ethical Concerns:**

["NO or VERY MINOR ethics concerns only"]

**Final Justification:**

Thank you for your response. The authors have addressed some of our concerns, we will keep our rating unchanged.

**Limitations:**

no
They claim to provide the limitations in section 2.1, but I don't find them.

**Quality:**

3

**Strengths And Weaknesses:**

**Strength**

This article uses post-neural architecture search to develop JetLM. It proposes a new linear attention module DynaConvGDN. Experimental results also show that JetLM achieves better results than Qwen2.5 on MMLU, Math Task and Commonsense Task, while significantly increasing throughput.

**Weaknesses**

Are the results achieved on MMLU, Math Task and Commonsense Task SOTA? Or are they only compared with the general LLM? Because the training set used by JetLM is not consistent with that of other general models, it is doubtful whether the final results can be attributed entirely to the DynaConvGDN and Hardware-Aware Architecture Search.

The performance of DynaConvGDN in Table 1 does not seem to be significantly better than Gated DeltaNet, and the paper does not provide experimental results on applying Gated DeltaNet to JetNet.

---

> ### Author Rebuttal · Authors · 2025-07-31
>
> ### Q1: Regarding Comparison Against SOTA Models
> We would like to clarify that JetLM is a general-purpose language model and has not undergone any domain-specific fine-tuning on downstream tasks such as MMLU, Math, or Commonsense. To ensure fair, apples-to-apples comparisons, we evaluate JetLM only against other state-of-the-art general-purpose language models. Comparisons with specialized or non-general-purpose models are beyond the scope of this work.
>
> ### Q2: Controlled Study to Exclude the Influence of Training Data
> We have conducted comprehensive controlled ablation studies (see Figure 3, Figure 5b, Table 1, and Table 2) to verify the effectiveness of each design component in JetLM. All models in these experiments were trained on the same dataset to ensure a fair comparison.
>
> Additionally, we fine-tuned the baseline models (Qwen2.5, RWKV-7, and Mamba-2) on JetLM’s training dataset to provide a more comprehensive evaluation.
>
> |         Model        | MMLU | Math | Commonsense | Retrieval |
> | --------------------- | ---- | ---- | ----------- | --------- |
> | Qwem2.5-1.5B-Finetune | 56.7 | 37.6 | 59.8        | 71.5      |
> | Mamba2-2.7B-Finetune  | 41.0 | 22.5 | 56.9        | 55.9      |
> | RWKV7-1.5B-Finetune   | 49.8 | 25.2 | 59.3        | 57.2      |
> | JetLM-2B              | **59.6** | **40.2** | **61.7**        | **73.6**      |
>
> JetLM-2B outperforms all these finetuned baseline models by a significant margin.
>
> ### Q3: Comparison Between DynaConvGDN and Gated DeltaNet
> We have included results for the JetLM variant that incorporates the Gated DeltaNet block in the manuscript. In Table 1, "Gated DeltaNet" refers to this JetLM variant, not the original standalone Gated DeltaNet model. We will revise the text in the next version to clarify this point.
>
> Beyond the MMLU, Math, Retrieval, and Commonsense tasks presented in Table 1, we further compare the DynaConvGDN block with the Gated DeltaNet block on four additional, more challenging benchmarks: MMLU-Pro, BBH, MBPP, and HumanEval.
>
> |                | MMLU-pro | BBH  | MBPP | HumanEval |
> | -------------- | -------- | ---- | ---- | --------- |
> | Gated Deltanet | 21.3     | 34.9 | 30.4 | 21.5      |
> | DynaDconvGDN   | **23.2**     | **37.6** | **33.4** | **24.4**      |
>
> The DynaConvGDN block outperforms the Gated DeltaNet block across all evaluated tasks, achieving an accuracy improvement of 1.9 to 3.0. These results clearly demonstrate the effectiveness of the DynaConvGDN design.
>
> Moreover, our contributions extend well beyond the DynaConvGDN block. As shown in Figure 3, the searched attention layer placement, linear attention selection, and hardware-aware architecture search each contribute significantly to overall accuracy gains. Additionally, our approach substantially reduces both the cost and risk of LLM architecture exploration, offering practical benefits to the broader language model community.
>
> ### Q4: Reasons for the Attention Layer Placement
> We determine the attention placement configuration based on our super-network search results, as detailed in Sections 2.2 and 3.1. Specifically, we use the Retrieval task to guide the placement of full attention layers and the MMLU task to guide the placement of sliding window attention (SWA) layers. As a result, we assign full attention to the top two critical layers for Retrieval (layers 15 and 20), and SWA to the top two critical layers for MMLU (layers 21 and 22).

---

> ### Author Response · Authors · 2025-08-04
>
> Dear Reviewer Ndha,
>
> We are truly grateful for your valuable feedback and the time you’ve taken to review our work. We would be honored to hear if our responses have satisfactorily addressed your concerns, and we warmly welcome any further suggestions you may have.
>
> With sincere thanks and best regards,
>
> The Authors

---

> ### Author Response · Authors · 2025-08-06
>
> Dear Reviewer Ndha,
>
> Thank you once again for your thoughtful suggestions and comments. As the discussion period deadline approaches, we would greatly appreciate it if you could let us know whether our responses have adequately addressed your concerns.
>
> Thank you!
>
> Sincerely,
>
> The Authors

---

> > ### Comment · Reviewer_Ndha · 2025-08-07
> >
> > Thank you for your response. The authors have addressed some of our concerns, we will keep our rating unchanged.

---

### Official Review · Reviewer_TeDN · 2025-07-04

**Clarity:** 2
**Significance:** 2
**Originality:** 2
**Rating:** 4
**Confidence:** 3

**Summary:**

This paper proposes a novel framework for searching efficient language models, which consists of finding optimal attention blocks, developing new blocks, and searching for the optimal number of attention blocks and their dimensions based on hardware constraints. Extensive experiments are performed on various datasets, showing that JetLM outperforms other methods in terms of throughput and accuracy.

**Questions:**

I have several questions:
1. Can the authors report the searching/training cost, and the final architecture ?
2. Which search algorithm is used in hardware-aware arch search? (RL/EA?)
3. How large is the search space used in Section 2.2 and 2.5? Can the author provide in detail? For example, range of $d_V$ and $d_K$?

**Ethical Concerns:**

["NO or VERY MINOR ethics concerns only"]

**Final Justification:**

Although the paper shows limited novelty from a NAS perspective, JetLM achieves fast inference speed for LLMs, which is a key factor for practical deployment. I therefore vote for borderline accept.

**Quality:**

2

**Strengths And Weaknesses:**

**Strengths:**
- The paper is easy to follow.
- JetLM runs faster than others while achieving better performance.

**Weaknesses:**
- While JetLM is effective, the proposed framework is not new in NAS community. As mentioned in related works, searching for optimal operations, number of layers, and hardware-aware search have been excessively studied in image classification and object detection tasks.
- Lack of reporting final architectures, searching/training cost.

---

> ### Author Rebuttal · Authors · 2025-07-31
>
> ### Q1: Novelty and Technical Contributions
> We respectfully disagree. The technical contributions presented in our work go well beyond the boundaries of the NAS community and address broader challenges in efficient language model design.
> + First, we introduce PostNAS, a novel model architecture exploration paradigm for language models. A key distinction between PostNAS and prior NAS methods for image classification or object detection is its ability to flexibly explore attention block designs while reusing pre-trained LLMs. In contrast, previous NAS approaches require training from scratch to achieve similar objectives, which renders them ineffective for large language models.
> + Second, our work offers novel insights into language model architecture design, such as the task-specific importance of attention layers and the finding that KV cache size is a more critical factor than parameter count for generation throughput.
> + Third, our work introduces a novel linear attention block, DynaConvGDN, which integrates linear attention with dynamic convolution and hardware-aware architecture search. It consistently delivers significant accuracy improvements over previous linear attention blocks while maintaining comparable generation throughput.
> + Fourth, our work introduces JetLM, a novel hybrid-architecture language model family that achieves superior accuracy and significantly higher generation throughput than prior state-of-the-art full-attention models (e.g., Qwen2.5, Gemma3, and Llama3.2). With its strong accuracy and exceptional inference efficiency, JetLM offers practical benefits for a wide range of applications requiring efficient language models.
> + Fifth, our work significantly reduces the cost and risk associated with LLM architecture exploration, enabling faster and more efficient innovation in language model design. This contribution can benefit the broader language model community.
>
>
>
> ### Q2: Final Model Architecture
> The final JetLM models are composed of a stack of blocks, each containing a Multi-Layer Perceptron (MLP) layer and an attention layer. The attention layer is selected from one of three types: full attention, sliding window attention, or DynaConvGDN. The detailed architecture configurations are presented below:
>
> |                                 | JetLM-2B   | JetLM-4B               |
> | ------------------------------- | ---------- | ---------------------- |
> | Total blocks                    | 28         | 36                     |
> | Full Attention Layers           | No. 15, 20 | No. 18, 21, 22,28, 33  |
> | Sliding Window Attention Layers | No. 21, 22 | No. 17, 20, 23, 24, 26 |
> | Vocabulary Size                 | 151643     | 151643                 |
> | Hidden Size                     | 1536       | 2048                   |
> | MLP Intermediate Size           | 8960       | 11008                  |
>
> The full attention and sliding window attention layers use grouped-query attention and are configured as follows:
>
> |      Full Attention / SWA    | JetLM-2B | JetLM-4B |
> | --------------------- | -------- | -------- |
> | Attention Head Number | 12       | 16       |
> | Dimensions of Q/K/V   | 128      | 128      |
> | K/V Head Number       | 2        | 2        |
> | Position Embedding    | RoPE     | RoPE     |
>
> For sliding window attention layers, the window size is set to 1,152 in JetLM-2B and 2,048 in JetLM-4B.
>
> The DynaConvGDN layers are configured as follows:
>
> |      DynaConvGDN    | JetLM-2B | JetLM-4B |
> | --------------------------- | -------- | -------- |
> | Q/K Dimension               | 96       | 128      |
> | V Dimension                 | 256      | 256      |
> | Head Number                 | 12       | 16       |
> | Convolution Kernel Size     | 4        | 4        |
> | DConv Generator Hidden Size | 32  | 32   |
>
> ### Q3: Search and Training Cost
> The search and training costs for JetLM are summarized below:
>
> |           |                        | Tokens (B) | ZFLOPs |  Time (H100 GPU Hours) |
> | --------- | ---------------------------------------- | ------ | -------- | -------- |
> | Search | Full Attention Placement and Elimination | 50 | 0.8    | 808    |
> |           | Linear Attention Block Selection         | 50       | 4.0    | 3120    |
> |           | New Attention Block Design               | 50       | 0.8    | 624      |
> |           | Harware-Aware Arch Search              | 50       | 7.2    | 5616    |
> | Training  | Stage1                                           | 50        | 0.8    | 624     |
> |           | Stage2                                                | 100      | 1.6    | 1344   |
>
> ### Q4: Search Algorithm in Hardware-Aware Architecture Search
> We employed grid search for the hardware-aware architecture search.
>
> ### Q5: Search Space
> In Section 2.2, the search space consists of all combinations of selecting 2 layers out of 28, resulting in a total size of $C_{28}^2$.
>
> In Section 2.5, the search space is detailed in Table 2, with $d_K \in [64, 96, 128, 192, 256]$ and $d_V \in [144, 192, 256, 288, 384, 576]$.

---

> > ### Author Response · Authors · 2025-08-04
> >
> > Dear Reviewer TeDN,
> >
> > We are truly grateful for your valuable feedback and the time you’ve taken to review our work. We would be honored to hear if our responses have satisfactorily addressed your concerns, and we warmly welcome any further suggestions you may have.
> >
> > With sincere thanks and best regards,
> >
> > The Authors

---

> > ### Comment · Reviewer_TeDN · 2025-08-06
> >
> > Thank you for addressing my concerns. Most of them have been resolved, and I will increase my score to 4.

---

> > > ### Author Response · Authors · 2025-08-06
> > >
> > > We’re glad to hear that our responses addressed your concerns. Thank you again for your thoughtful comments!

---

### Decision · Program_Chairs · 2025-09-17

**Decision:**

Accept (poster)

**Comment:**

This paper proposes JETLM a method to select, drop or develop novel attention blocks and placement in a language model to speed it up. Interestingly this leads to some new attention mechanisms to be developed during the postNAS stage. All reviewers agree that although this is not a particularly novel approach, the advantages in speedup are noticeable. Also, the paper's writing has been appreciated.

I believe that this work should have practical applications and deserves a spot in the conference if theres space.